# Effect of Dried Leaves of *Leucaena leucocephala* on Rumen Fermentation, Rumen Microbial Population, and Enteric Methane Production in Crossbred Heifers

**DOI:** 10.3390/ani10020300

**Published:** 2020-02-13

**Authors:** María Denisse Montoya-Flores, Isabel Cristina Molina-Botero, Jacobo Arango, José Luis Romano-Muñoz, Francisco Javier Solorio-Sánchez, Carlos Fernando Aguilar-Pérez, Juan Carlos Ku-Vera

**Affiliations:** 1Faculty of Veterinary Medicine and Animal Science, Autonomous University of Yucatan, Mérida 97300, Mexico; 2National Center for Disciplinary Research in Physiology and Animal Breeding, National Institute for Forestry, Crops, and Livestock Research – Ministry of Agriculture and Rural Development, Ajuchitlán 76280, Mexico; 3The Alliance of Bioversity International and the International Center for Tropical Agriculture (CIAT), Km 17 Recta Cali-Palmira, A.A. 6713 Cali, Colombia

**Keywords:** *Leucaena leucocephala*, methane, digestibility, volatile fatty acids

## Abstract

**Simple Summary:**

The objective of the experiment was to evaluate the effects of dietary supplementation of heifers with increasing levels of dried *Leucaena*
*leucocephala* leaves (DLL) on nutrient digestibility, fermentation parameters, microbial rumen population, and production of enteric methane (CH_4_). Nutrient digestibility decreased with increasing levels of DLL in the ration. Inclusion of DLL did not have detrimental effects on rumen pH, rumen microbial community, and volatile fatty acids proportions. Enteric CH_4_ emissions in heifers were reduced with increasing levels of DLL in the ration.

**Abstract:**

The effects of dietary inclusion of dried *Leucaena*
*leucocephala* leaves (DLL) on nutrient digestibility, fermentation parameters, microbial rumen population, and production of enteric methane (CH_4_) in crossbred heifers were evaluated. Four heifers were used in a 4 × 4 Latin square design consisting of four periods and four levels of inclusion of DLL: 0%, 12%, 24%, and 36% of dry matter (DM) intake. Results showed that DM intake (DMI), organic matter intake, and gross energy intake (GEI) were similar (*p* > 0.05) among treatments. Apparent digestibility of organic matter, neutral detergent fiber, and energy decreased with increasing levels of DLL in the ration (*p* < 0.05). In contrast, digestible crude protein (CP) was higher (*p* < 0.05) in treatments with 12% and 24% DM of DLL. The inclusion of DLL did not affect (*p* > 0.05) rumen pH and total volatile fatty acids. Rumen microbial community was not affected (*p* > 0.05) by treatment. There was a linear reduction (*p* < 0.05) in CH_4_ emissions as the levels of DLL in the ration were increased. Results of this study suggest that an inclusion of 12% DM of ration as DLL enhances digestible CP and reduces daily production of enteric CH_4_ without adversely affecting DMI, rumen microbial population, and fermentation parameters.

## 1. Introduction

Methane (CH_4_) is the second most important greenhouse gas (GHG) and it has a potential global warming effect 28 times above that of carbon dioxide (CO_2_) [1]. The livestock sector contributes 14.5% of global emissions of GHG [2] and CH_4_ represents 44% of total anthropogenic emissions [3], where CH_4_ from ruminant enteric fermentation represents 39.1% of total emissions in the livestock sector [2]. Methane is a byproduct of anaerobic microbial fermentation of feed in the rumen, and energy used for its synthesis is considered as a loss of energy for animal production; it has been calculated that the energy loss fluctuates between 3% and 6.5% on average for cattle fed diets high in concentrates and low-quality pastures, respectively [4]. In tropical climates, feeding of ruminants is largely sustained by using low-quality forages, which in turn increases the production of CH_4_ [5]. Facing the current climate issues, research has been focused on the reduction of enteric CH_4_ through feeding practices that alter rumen fermentation, as well as the use of mitigating agents, such as essential oils [6], secondary metabolites [7,8,9], and some chemical compounds such as organophosphates [6,10] or 3-nitrooxypropanol [11]. In tropical climates, *Leucaena leucocephala* is a legume species that is highly available and commonly used as fodder for ruminant feeding. Incorporation of *L*. *leucocephala* in tropical pastoral systems for meat production has proven to reduce CH_4_ emissions in cattle [12,13]. The use of this tropical legume in ruminant nutrition has been widely implemented due to its high content of crude protein. On the other hand, the effect of *L. leucocephala* on enteric CH_4_ reduction is linked to its content of condensed tannins (CT), which form complexes with protein (CT–P) and with polysaccharides, and reduce nutrient degradation in the rumen [14,15]. In addition, some studies propose that CT promote changes in microbial populations [16,17,18] due to bacteriostatic, bactericidal, and enzyme inhibiting effects that modify rumen fermentation.

The aim of the study was to quantify the effect of increasing levels of dried *L. leucocephala* leaves on nutrient intake and digestibility, rumen fermentation patterns, CH_4_ production, and rumen microbial populations in crossbred heifers.

## 2. Materials and Methods

### 2.1. Animals, Diet Management, and Experimental Design

The study was conducted at the Laboratory of Climate Change and Livestock Production at the Faculty of Veterinary Medicine and Animal Science (FMVZ-UADY) of the Autonomous University of Yucatan, Merida, Mexico. Management of experimental animals followed the protocol for animal guidelines and regulations for animal experimentation and welfare of FMVZ-UADY and the experimental protocol was conducted in accordance with the Mexican Official Standard NOM-062-ZOO-1999, technical specifications for the production, care, and use of laboratory animals. Four crossbred heifers (*Bos taurus* × *Bos indicus*) with an average body weight (BW) of 310 ± 9.6 kg (mean ± SD) were used. Before the experiment started, heifers were dewormed with Ivermectin^®^ (Pier; Dose: 1 mL/50 kg BW) and injected intramuscularly with vitamins A, D and E (Vigantol ADE^®^, Bayer manufacturer, Köln, North Rhine-Westphalia, Germany). The heifers were accustomed to the indirect open-circuit respiration chambers for CH_4_ measurements before starting the experiment. Drinking water was available ad libitum. Heifers were randomly assigned in a 4 × 4 Latin square design with four treatments, four heifers, and four periods. Each period lasted for 15 days: 1 to 8 for treatment adaptation and days 9 to 15 for measurements. In order to minimize the residual effect of treatments, after every period, heifers were fed with a diet without dried *Leucaena leucocephala* leaves (DLL) for 10 days (cleansing). 

### 2.2. Experimental Diets

Young stems of leaves of *L. leucocephala* at 45 days of growth were harvested, the leaves were chopped and dried in the shade for 4 days, and then oven dried in a stove at 40 °C for four days. Dried *L. leucocephala* leaves were stored and protected from light until used. Nutrient diet formulation was based on metabolizable energy and protein requirements for ruminants according to the Agriculture and Food Research Council [19], for a predicted daily weight gain of 0.75 kg. Diet formulation was aimed to be isoenergetic, isoproteic, and with similar content of neutral detergent fiber (NDF). Diets were offered to the heifers as total mixed ration, in order to maintain homogeneity in particle size and particle type among rations. Formulation and chemical composition of each experimental diet are shown in Table 1. The inclusions of 0%, 12%, 24%, and 36% dry matter (DM) per animal^−1^ day^−1^ of DLL corresponded to treatments 0, 1, 2, and 3. Experimental diets were offered at 08:00, considering a dry matter intake (DMI) of 2.5% of BW [20]. The feed offered was adjusted based on the BW of each animal for every period. Heifers were weighed two days before the beginning of every period and two days after finishing every period.

Voluntary intake of DM and nutrients in experimental diets were determined as the difference between the amount of nutrients offered and the amount which was refused every day. Samples of feed and refusals were collected and stored every day for posterior chemical analysis. Apparent digestibility was determined by using the method described by [21]. Total production of feces was collected and weighed every day, and an aliquot of 10% was stored for further analysis. Fecal sample aliquots (last six days during every period) were pooled for treatment each period and were used for chemical analysis.

Estimated nonfiber carbohydrate (NFC) and total digestible nutrients (TDN) were calculated according to Nutrient Requirements of Dairy Cattle [22]. Metabolizable energy (ME) concentration was calculated considering that 1 kg of TDN is equal to 4.409 Mcal of DE and 1 Mcal of DE is equal to 0.82 Mcal of ME [22].

### 2.3. Chemical Analysis

The collected samples of feed and orts as well as feces were ground and passed through a 1 mm sieve for analysis according to the methods [23]. DM content of diet, orts, and feces were determined by drying subsamples in a forced-air oven at 105 °C for 48 h (constant weight). Nitrogen (N) and crude protein were carried out with Kjeldahl AN 3001 FOSS [23]. Crude protein was calculated as N × 6.25. Organic matter (OM) content of the samples was determined by combustion in a muffle furnace at 550 °C [23] and the concentration of NDF (AN 3805 ANKOM, ANKOM Technology, Wayne County, NY, USA) [23] and acid detergent fiber (ADF) (AN 3804 ANKOM) [23] were determined as suggested by Van Soest et al. [24]. Gross energy (GE) was measured using a bomb calorimeter (6400 Parr Instrument Company, Moline, IL, USA). Acid detergent lignin (ADL) was determined in beakers (ANKOM Technology, Wayne County, NY, USA). Ether extract (EE) contents were obtained by the Randall method (SER 148 Solvent extraction unit) [23]. In vitro DM digestibility was determined as suggested by Tilley and Terry [25]. Total phenol (TP) content and total tannins (TT) were determined following the Folin–Ciocalteu method; precipitating tannins with the polyvinylpolypyrrolidone [26], and were expressed as acid-tannin equivalents g kg^−1^ DM. CT were quantified using the vanillin assay [27], and were expressed as vanillin equivalents in g kg^−1^ DM. 

### 2.4. Blood Samples

Blood samples were collected by jugular venepuncture using 7-mL blood collection tubes (Vacutainer; BD Inc., Oxford, UK) on days 14 and 15 for every period, within four hours postprandial. Blood samples were immediately centrifuged (2500 rpm for 10 min at 4 °C) to separate serum, which was stored at −20 °C until further analysis. Blood urea nitrogen was determined via a colorimetric assay using a commercial kit (Accutrack S.A. de C.V., CDMX, Mexico).

### 2.5. Rumen Fermentation

Samples of rumen content were taken from animals on days 14 and 15 of every period, within four hours postprandial. Approximately 1.2 L of rumen fluid was taken from each heifer by inserting an esophageal tube [28]. The samples were filtered through four layers of cheesecloth. Three sub-samples were taken. Metaphosphoric acid was added to the first sub-sample 1:5 (v/v) to preserve the sample at −80 °C for volatile fatty acids (VFA) analysis. VFA proportions in rumen liquid were determined by gas chromatography (7890A GC system Agilent Technologies Inc, Santa Clara, CA, USA), equipped with a flame ionization detector [29]. Rumen pH was measured immediately after obtaining the sample with a portable potentiometer (HANNA^®^ Instruments, Woonsocket, RI, USA) in the second sub-sample. The third subsample was preserved at −80 °C for ruminal microbial deoxyribonucleic acid (DNA) extraction.

### 2.6. Microbial Quantification

#### 2.6.1. Ruminal Microbial DNA Extraction

The deoxyribonucleic acid extraction was carried out using the method described by Rojas-Herrera [30]. DNA concentration was calculated using a NanoDrop 2000 (Thermo Scientific, Waltham, MA, USA), and the DNA integrity was confirmed by agarose gel electrophoresis. DNA samples were then stored at −80 °C until analysis.

#### 2.6.2. Quantitative Real-Time Polymerase Chain Reaction (qPCR)

The qPCR was applied to quantify the bacterial, protozoal, and methanogenic archaeal populations in the rumen by measuring the absolute quantity of the targeted DNA fragments by a reference to a standard curve constructed with a plasmid containing the target insert. The domain bacteria primers used were BAC338F and BAC805R [31], methanogen primers were Met630F and Met803R [32], and protozoal primers were Oph-151F and Ento-472R [33] (Table 2). The qPCR amplifications for the quantification of target ruminal microbial genes were performed using a Rotor-Gene Q (Qiagen, Hilden, Germany). Chemical reagent Go Taq Green Master Mix (Promega, Madison, WI, USA) was used following the manufacturer’s instructions. 

Each gene was cloned separately by using pGEM^®^-T Easy Vector System and ligation using 2× Rapid Ligation buffer (Promega, Madison, WI, USA) according to the manufacturer’s instructions. Then, the recombinant vector was transformed into competent *E. coli* cells with ampicillin and X-gal/IPTG. Transformed positive colonies were picked and processed for plasmid isolation. Plasmid purification was done using a Wizard^®^Plus SV Minipreps DNA Purification System (Promega, Madison, WI, USA.). Presence of the insert in the recombinant clones was confirmed by restriction digestion and digested products were detected by agarose gel electrophoresis. Plasmid DNA concentrations were measured using a NanoDrop 2000 (Thermo Scientific, Waltham, MA, USA) and copy numbers were calculated using the following equation:
(1)$$\mathrm{Number}\;\mathrm{of}\;\mathrm{copies}/µL=\frac{6.022\times \;10^{23}\;\left(\frac{\mathrm{molecules}}{\mathrm{mole}}\right)*plasmid\;concentrations\;\left(\frac{g}{µL}\right)}{number\;of\;bases*660\;daltons}$$

To construct the standard curve, a tenfold series dilution (10^2^ to 10^10^) was performed for each target gene. Each standard curve and sample was analyzed in triplicate for absolute quantification. The amplification efficiency (*E*) was determined by the slope (*s*) of linear regression of the standard curve. Amplification efficiency was established using the following equation [34]: (2)$$E=10^{\left(-1/s\right)}$$

### 2.7. Methane Production

Emission of CH_4_ was measured using individual indirect open-circuit respiration chambers [35] with a methane measurement system (Sable Systems International^®^, Las Vegas, NV, USA). Heifers remained in the respiration chambers for 23 h per day. Each heifer was in the chamber for four days receiving the same treatment during the period. Temperature and relative humidity inside the chambers were kept at 23 °C and 55%, respectively. Data were extrapolated to 24 h using Expe Data^®^ software (Sable Systems International^®^, Las Vegas, NV, USA). The infrared CH_4_ analyzer was calibrated before each experimental period. Methane concentrations were transformed to energy taking into account the heat combustion of CH_4_ (55.65 MJ kg^−1^) [4]. Moreover, methane loss was expressed in absolute terms as GE (Gross energy, MJ d^−1^) and also emission of CH4 as a percentage of GEI (Gross energy intake, Ym). Additionally, the estimate of the emission factor (EF) of CH_4_ kg animal^−1^ year^−1^ was calculated according to [4].

### 2.8. Nitrous Oxide Emissions

Estimated nitrous oxide (N_2_O) emissions from feces of heifers were calculated with fecal N excretion and according to its volatilization rate for Latin America [4] with the equations 10.25 (direct N_2_O emissions from manure management), 10.28 (N losses due to leaching from manure management system), and 10.29 (indirect N_2_O emissions due to leaching from manure management) of the [4].

Both CH_4_ production and estimated N_2_O emissions were converted to CO_2_ equivalents (CO_2_-eq) using global warming potential values of 28 and 265, respectively [4].

### 2.9. Statistical Analyses

Statistical analyses were carried out on data of DM intake, apparent digestibility, fermentation parameters, blood metabolites, CH_4_ production, and rumen microorganism population. Data of bacteria, methanogen, and protozoa (cells mL^−1^ of ruminal liquor) were transformed to Log_10_. All data were subjected to analysis of variance for a 4 × 4 Latin square design, using the mixed procedure of the SAS^®^ 9. Software (SAS Inc., Cary, NC, USA.) [36]. The statistical model was Yijk = μ + Pi + Aj + Tk + Eijk; where: Y is the dependent variable, μ is the general mean, P is the effect of period, A is the random effect of animal, T is the effect of treatment, and E is the random residual error. Results were compared with the procedure LSmeans test, whereas orthogonal contrasts were performed to evaluate the effect of treatments [36].

## 3. Results

### 3.1. Chemical Analysis of Experimental Diets

The chemical analyses of experimental diets are shown in Table 3. Although the aim of diet formulation was to obtain isoproteic and isoenergetic diets, it was evident that the diet without inclusion of DLL had a lower content of CP and GE in comparison with the other treatments. In addition, the diets were not homogeneous in the contribution of NDF. In the case of ADF and ADL, these were increased as a result of increasing the inclusion of DLL in the diets. The concentration of ADF and ADL in treatment with DLL showed a trend higher than treatment 0. These could be related to a minor content of OM and reduction of in vitro DM digestibility. On the other hand, the concentrations of fat and secondary metabolites (total phenols, total tannins, and CT) in the diets were increased as a result of a higher level of inclusion of DLL in diets. Estimation of NFC showed the lowest content in treatment 3 compared to the other treatments.

### 3.2. Feed Intake and Apparent Digestibility 

The effect of DLL supplementation on intake and apparent digestibility is presented in Table 4. No significant effects were found in the DMI, OM intake (OMI), and GEI. On the other hand, the intakes of CP and ADF increased with a higher level of DLL. Crude protein intake was 21.7%, 28.3%, and 28.3% higher for treatments 1, 2, and 3 than that for treatment 0. In contrast, the consumption of NDF was lower by 5% for treatments 1 and 2 compared to treatments 0 and 3. Consumption of ADL was different between treatments 1 and 2. Intake of TP, TT, and CT increased linearly as DLL inclusion increased. For CT, an increase corresponding to 2.7, 8.2, and 12.3 g kg^−1^ of DMI for treatments at 1, 2, and 3 was observed compared with treatment 0. Apparent digestibility of OM and NDF decreased with increasing DLL in the rations. The effect of reduction in digestible OM and NDF were of 1.5%, 8.8%, 13.2% and 12.7%, 22.5%, 13.4% for treatments 1, 2, and 3, respectively against treatment 0. On the other hand, digestible energy (DE) decreased to 8.3 MJ d^−1^ with treatment 3 versus treatment 0. In contrast, the apparent digestibility of CP increased compared to the control by 25% and 14% for treatment 1 and 2. In the same way, the digestibility of ADF increased by 20% in treatment 2 in contrast to treatment 0.

### 3.3. Fermentation Parameters

Rumen fermentation parameters are shown in Table 5. Ruminal pH was not affected (*p* > 0.05) by the inclusion of DLL in the diets. Molar concentrations and the ratio of acetic acids in rumen liquor were affected significantly by treatments (*p* < 0.05).

### 3.4. Fecal Nitrogen Excretion

The values of ingested and fecal excreted nitrogen (N) are shown in Table 6. Ingested N (Ni) show differences (*p* < 0.05) among treatments. However, the concentration of blood nitrogen urea (BUN) increased (*p* < 0.05) in treatments 1 and 2 compared to the other treatments, while fecal N excretion (Nf) increased linearly (*p* < 0.0001) as DLL increased in the diet.

### 3.5. Ruminal Microorganism Population 

The effect of supplementation of DLL on ruminal microorganism populations is summarized in Table 7. Inclusion of DLL showed not effect (*p* > 0.05) on protozoal, bacterial, and archaeal populations. The ratio of methanogens:bacterial population was not altered (*p* > 0.05). 

### 3.6. Methane Production.

Methane production is shown in Table 8. Inclusion of DLL in the diets resulted in a significant decrease (*p* < 0.001) in CH_4_ production. Methane reductions were of 6.2%, 11%, and 19.6% (where 100% is equivalent to the emission of CH_4_ measured without the inclusion of DLL or treatment 0) corresponding to an increase of DLL in the diet. In this study, the production of CH_4_ g kg^−1^ DMI was affected negatively (*p* < 0.001) in treatments 2 and 3, which corresponds to a reduction of 13.5% and 20.7%, respectively. Reduction of CH_4_ g kg^−1^ of DM, CH_4_ g kg^−1^ of digestible OM and CH_4_ g kg^−1^ of digestible NDF did not show significant differences (*p* > 0.05) among treatments. In contrast, CH_4_ g kg^−1^ of digestible CP decreased significantly (*p* < 0.001) by 25 % on average in all treatments containing DLL against treatment 0.

Energy loss as CH_4_ during fermentation was 6.2%, 11.3%, and 19.6% MJ d^−1^ of GEI for treatments 1, 2, and 3, respectively versus treatment 0. These effects showed a linear (*p* < 0.01) reduction of energy correlated negatively with DLL inclusion. In the same case, Ym decreased (*p* < 0.01) by 6.6%, 14.8%, and 22.3 % in treatments 1, 2, and 3, respectively versus treatment 0. The emission factor (EF) of CH_4_ kg animal^−1^ year^−1^ decreased linearly (*p* < 0.01) in response to inclusion of DLL. Treatments 1, 2, and 3 reduced EF by 6.5%, 11.1%, and 19.6% in contrast to treatment 0.

### 3.7. Effect of DLL on Greenhouse Gas Emissions

Estimated global warming potential (GWP) of CH_4_ and N_2_O in heifers fed DLL can be observed in Table 9. Estimated GWP from CH_4_ as CO_2_-eq d^−1^ showed a decreasing tendency as DLL increased in the diets. In contrast, all treatments with DLL showed an increase of estimated GWP through N_2_O CO_2_-eq d^−1^ from nitrogen excreted through feces against the treatment without DLL. However, according to the estimate of total GWP (CH_4_ and N_2_O) all treatments with DLL showed a total GHG emission mitigation potential compared to the diet without DLL or treatment 0.

## 4. Discussion

The effects of *L. leucocephala* consumption on intake and digestibility of feed, and on enteric CH_4_ mitigation, are linked to its content of CT. Condensed tannins have been widely studied for their effects in animal nutrition. It is generally accepted to use the term ‘anti-nutritional effect’ to describe the reduction of palatability leading to a reduced ingestion of feed and a lower nutrient digestibility. Both effects are related to the astringency generated by the capacity of tannins to establish stable bonds with dietary components [37]. In this study, the DLL treatments contained 0%, 0.28%, 0.82%, and 1.23% CT in the total diet. None of the inclusion levels of *L. leucocephala* affected DMI or OMI. Previous studies obtained similar results [8,9], but did not observe any effect on DMI with CT doses ≤2%. Ruminants in the tropics are exposed to tannin containing forages, leading to selectivity in their consumption or adaption to these conditions. Some studies have suggested that ruminants possess proteins with a high content of amino acids such as proline [38] in their saliva, which are more likely to bind with CT. Thus, allowing ruminants to reduce or block the effect of astringency which could lead to a reduction in feed intake. 

In regard to CP and ADF consumption, significant increments were shown with DLL supplementation which was related to the concentration of these components in the diet (Table 3). In the case of legumes such as *L. leucocephala* there has been reported greater amount of CP compared to tropical grass [39]. On the other hand, NDF and ADL intakes remained unchanged or decreased compared to those for treatment 0. This observation is contrary to previous studies where no effect on consumption of NDF and ADF with doses ≤2% CT was observed [8,40].

Apparent nutrient digestibility was affected by the level of DLL in the ration. Digestibility of OM and NDF showed a linear reduction. These findings may be associated with in vitro DM digestibility (Table 4) and total digestible nutrients (TDN) which showed a comparable decline with higher inclusions of DLL. This effect can be explained by the ADL contained in DLL. Similar reductive effects were reported in other studies for digestibility of DM (DDM) [41,42], digestible OM (DOM) [39], and digestible NDF [42]. However, results reported by other authors differ from the above, showing increases in DOM, NDF, and digestible ADF [8]. It also indicates increases in digestibility of OM and NDF [43]. All results quoted in this section were derived from in vivo studies and correspond to a dose of ≤2% CT. However, the source of CT differs in some of the studies, which could account for the heterogeneity of results. Genotype, species, variety, and growth stage of the plant material are associated with differences in the chemical structure and molecular weight of CT as key characteristics determining their capacity to precipitate proteins [41,44] which could affect the digestibility of feed. Another aspect that has to be taken into account to explain the differences regarding digestibility is the ability of CT to attach to cellulose and hemicelluloses as well as to enzymes of microbial origin [17,45]. Consequently, it makes sense to expect a decrease in digestibility of the fibrous fractions, DDM, and DOM as a result of the inclusion of CT [46] in the ration.

In this study, treatment 3 (1.2% CT in diet) reduced DE significantly (Table 2). However, that probably did not result from CT contained in diet, because estimation of metabolizable energy (ME) also showed a reduction derived from low TDN concentration in treatment 3. Similar reductive effects in DE have been reported elsewhere [9].

Digestibility of CP was significantly higher for treatments 2 and 3 relative to treatment 0 (Table 2). Similar results have been found for doses of 0.83%, 1.37%, and 1.89% CT [47]. Results suggest that there were effective bonds between dietary proteins and CT in the rumen and consequently the intestinal absorption of protein was increased [37,43]. However, in this study the dose of 1.2% CT (treatment 3) did not increase digestible CP. This result for treatment 3 can be partially explained by the disproportion between the contributions of energy and protein that affects microbial fermentation in the rumen. Several studies are in agreement showing a linear digestible CP depression at increased doses of CT [8,9,41,42].

Several authors have attempted to explain the reduction of digestibility by an incomplete dissociation of the CT-P complexes in the abomasum (due to pH conditions) reaching the intestine and not being absorbable. In this case, the synthesis of irreversible bonds between CT and dietary proteins is suggested. However, there is evidence that dietary CT-P bonds are reversible and largely dissociated in the abomasum [48]. Other authors propose that the digestibility reduction results from the ability of CT to form new complexes with N from endogenous origin within the intestine [17,48]. This theory has been supported by several studies involving N balance [9,41,42].

The hydrogen potential is a quantitative measure of acidity or alkalinity of a solution. A stable rumen pH is a precondition for the growth of microorganisms, the fermentation of ingested feed, and the absorption of organic acids [49]. In the present study rumen pH values did not vary significantly (on average 6.5 ± 0.05) among treatments. Similar results have been reported elsewhere, with average pH values of 6.6 ± 0.2 at CT concentrations of <2% [40,41,42,47]. This result could be explained by the proportion of forage in the rations of the above mentioned studies, which was kept at above >50% of DM. Forage consumption stimulates salivary excretion which in turn is an important factor in maintaining a stable rumen pH [49].

In this study, CT from DLL did not affect rumen fermentation with regard to total VFA concentration, molar proportions of propionate (C_3_), butyrate (C_4_), and the acetate to propionate ratio C_2_:C_3_. These results are in accordance with other studies [40,41,42], all of them at doses ≤2% CT. On the other hand, they differ from results showing a reduction in total VFA concentration [8], C_2_ [8], C_4_ [47], and C_2_:C_3_ ratio [8,47]. Furthermore, increments of C_3_ [47] have been reported. The difference of effects on rumen fermentation might be related to the doses and ability of CT in forming complexes with dietary proteins, inhibition of catalytic activity of extracellular enzymes, or the reduction of bacterial populations [17]. It has also been proposed that CT reduces the acetic acid:propionic acid ratio which in turn reduces the amount of available hydrogen for methanogenesis [50]; however, in this study such effect was not observed.

The CP:TDN ratio of 0.19 or 190 g CP kg^−1^ TDN is a reference for efficient daily weight gain (0.84 kg) and minimal CH_4_ emissions for cattle during growth compared to cattle fed at lower CP:TDN values [51]. Although the concentration of nitrogen intake was similar among treatments with DLL, the CP:TDN ratio (Table 6) indicates with higher precision the differences in nitrogen intake as a result of increasing DLL in diets. Furthermore, the ratio of TDN:CP can be used to identify the balance of nutrients in the rumen. A TDN:CP ratio between 4:1 to 7:1 is considered an adequate contribution of N in cattle. Ratios higher than 7:1 indicate a deficiency of rumen degradable CP. On the other side, ratios that are <4:1 show an excess N or a lack of energy relative to the amount of rumen degradable CP [52]. In this study, following the criterion of TDN:CP, treatments 2 and 3 suggest an excess CP.

Concentration of BUN correlates directly with the concentration of CP in the feed and the concentration of ammonia in the rumen. BUN range for cattle has been established as an indicator for desirable productive performance. It varies depending on the productive phase and the production system, in the case of growing animals, a range between 9 and 12 mg dL^−1^ has been suggested as optimal [53]. BUN concentrations outside this general range in cattle are indicative of deficiency (<6 mg dL^−1^) and excess of protein supply (>19 mg dL^−1^) [53,54]. BUN determined for the treatments hereby described are within the range for growing cattle. Treatments 1 and 2 showed a higher BUN compared to treatment 0. In accordance with these results in another study, doses of 0.83% and 1.89% CT increased BUN compared to the control treatment [47]. By contrast, a reduction of BUN at doses below 1.9% CT has been also reported [42], which compares well with the reduction in BUN observed at the dose of 1.2% CT in this trial. These findings may be associated with the capacity of CT–P complex formation which in turn depends on the source and the growth stage of the legume [41,44].

Results previously shown, indicated a reduced CP digestibility, which could be a consequence of increased complex formation between endogenous nitrogen and CT in the small intestine, which is then excreted in the feces. Thus, it seems important to explore in more detail the effects of such endogenous N excretion in feces. The loss of N in feces showed a linear positive trend, possibly correlated with CT concentrations in the diet. Previous studies revealed decreased CP digestibility at doses <1.9% CT, while N retention was not affected, the excretion of N in feces increased, whilst N excretion in urine was reduced [9,41,42]. On the other hand, the Nf:Ni ratio helps to explain effects related to the proportion of excreted Nf. In this case, treatments 2 and 3 showed increased Nf compared to the other treatments. This effect can be explained by the increase in CP and CT with high doses of DLL. This fact can be reinforced with PC:DOM ratio >0.288 for treatments 2 and 3 because the interval of 0.191–0.218 of CP:DOM ratio is interpreted as the range of higher efficiency of nitrogen utilization in grazing cattle [55]. Also, CP:DOM ratios greater than 0.288 suggest a loss of N [55].

Rumen populations of microorganisms are responsible for the fermentation of feed. As a result, most of the proposed strategy to reduce enteric CH_4_ production is by regulating their growth, quantity, and metabolism. Rumen microbiota (protozoa, bacteria, and methanogenic archaea) quantified by means of qPCR were not affected by treatments in this study. Previous investigations had obtained the same results of quantification analysis of rumen microbial populations [41,56]. One possible explanation is that some microorganisms are adapted with protective mechanisms against CT, such as the production of polymers for cellular protection and tannin degrading enzymes [17,50].

Results obtained did not show any difference with regard to the total quantity of microbial populations. However, the results on digestibility can be a consequence of increased inhibition of bacterial enzymatic activity with increasing CT in the diets. Some studies that include the identification of specific genera for microbial diversity and bacterial enzymatic activity using omic tools have shown in more detail that the effect of some secondary metabolites in rumen microorganisms are still present, while the quantity of microorganisms does not seem to be affected [57,58,59]. Enteric CH_4_ mitigation is undoubtedly related to direct and indirect effects of CT on microbial populations. The mechanisms of action of CT shown in several experiments are as follows: antimicrobial effect on cellulolytic and proteolytic bacteria, interference with the catalytic activity of extracellular enzymes acting on fermentation of feed, reduced availability and digestibility of nutrients [17,50]. Otherwise, CT induce a defaunating effect of protozoa [7,45,60]. In the case of methanogenic archaea, a growth inhibiting effect has been proposed [61]. 

In the present study treatments 2 and 3 (0.8% and 1.2% CT) decreased production of CH_4_ g kg^−1^ DM (13.5% and 20.7% compared to treatment 0) (Table 6). These results are similar with what was observed at 1.37% and 1.89% CT in DLL [47], and doses of 0.9% and 1.36% CT in *Acacia mearnsii* [9]. Reduction of enteric CH_4_ with the inclusion of DLL in the diets could be explained by the lower digestibility of the crude protein and OM in the rumen. This effect is attributed to the ability of the CT to form complexes CT-P [9,47].

The evaluation index of strategies (chemical products, ingredients, secondary metabolites, and feed management, amongst others) proposed in various studies for the mitigation of enteric CH_4_ emissions is usually expressed in g kg^−1^ DMI. However, it has been shown that the results obtained with this index differ from those defined as CH_4_ units per unit of product generated by cattle (milk or weight gain). The objective of the latter is to assess if the mitigation effect of CH_4_ compromises animal production [62,63]. As a consequence, more studies that include production variables are needed to discern the real potential of the strategies under investigation. In this context, the inclusion of doses of 0.9% and 1.36% CT in *Acacia mearnsii* decreased the production of CH_4_ [9]. However, the adequate concentration of CT recommended to reduce CH_4_ (14%) without negative effects on milk production was 0.9% [9].

On the other hand, the Intergovernmental Panel on Climate Change, (IPCC) [1,4] suggest to determine emissions of CH_4_ in terms of unity of GEI as Ym or as GE loss MJ d^−1^. In this study, all treatments with inclusion of DLL showed a reduction in enteric CH_4_ emissions when expressed both ways. For growing cattle (on pasture or in high fiber diets) Ym average value is 6.5% of GEI in Latin America [4]. The treatment without inclusion of DLL in the present study agrees with this reference.

GHG affect the atmosphere in different proportions and remains there for different lengths of time. The GWP evaluates GHG in relation to their warming potential of one unit of CO_2_ during the same period of time [4]. CH_4_ and N_2_O possess a GWP of 28 and 265 times higher than that of CO_2_, respectively [1].

The effect of CT in reducing enteric CH_4_ production has been widely documented [8,9,17,18,64]. However, few studies evaluate the effect of CT on N_2_O emissions. CT have been shown to increase the concentration of nitrogen in feces as a side effect [9,41,42]. The inclusion of the dose of 0.45% CT in the diet gave a reduction in the excretion of N in urine without affecting digestibility of nutrients and milk production [42]. It has also been mentioned, that N excreted through feces compared to N from urine is less volatile and as a consequence the N that can be converted to N_2_O is reduced [65,66]. Otherwise, N in feces bound to CT is more resistant to degradation under soil conditions compared to organic N in feces [67].

In this study, estimated N_2_O as CO_2_-eq d^−1^ in feces was increased due to higher CT concentration. On the other hand, enteric CH_4_ as CO_2_-eq d^−1^ showed a linear trend towards a reduction at increasing DLL concentrations in the diet.

In summary, the lowest total GWP in CO_2_-eq d^−1^ of gases was established for treatment 3 (1.2% CT in diet). Even though this treatment showed a lower digestible CP and DE, and also reduced BUN in contrast to the other doses of DLL. On the other hand, treatments 1 and 2 showed equal reductions in GWP compared to treatment 0. However, treatment 2 decreased digestibility of OM and NDF while increased the digestibility of ADF and excess of N compared to treatment 1.

This study did not include analysis of N excreted in the urine as variable where the largest production of N_2_O may have been expected. Hence, for future studies it is important to include evaluations of N balance and production variables with the aim of establishing the precise doses at which CT are most effective in mitigating GHG without affecting animal production. Similar observations have been suggested when evaluating mitigation strategies for extended periods of time and interpreting the results of mitigating GHG in kg of product (milk and meat) obtained [68].

## 5. Conclusions

The results of this experiment demonstrate that the inclusion of DLL in the ration of growing crossbred heifers decreased the emissions of enteric CH_4_. Supplementation of 12% of ration DM with DLL (0.27% CT in ration) was enough to increase the digestibility of dietary protein and organic matter while reducing CH_4_ production, without negative effects on the quantity of microbial populations and rumen fermentation.

## Figures and Tables

**Table 1 animals-10-00300-t001:** Ingredient composition of experimental diets.

Item	Treatments			
	0	1	2	3
Ingredients (g kg^−1^ DM)				
Guinea grass hay, ground	433.3	408.0	429.3	425.3
Corn grain, cracked	93.3	94.7	93.3	94.7
Soybean meal	66.7	66.7	37.3	0.00
*Leucaena leucocephala* dried leaves	0.00	120.0	240.0	360.0
Wheat bran	266.7	213.2	90.7	0.00
Sugarcane molasses	133.3	90.7	102.7	113.3
Mineralized salt	6.7	6.7	6.7	6.7
DM: dry matter				

Data collection: sampling procedures and analysis; dry matter intake and apparent digestibility.

**Table 2 animals-10-00300-t002:** Oligonucleotide primers used for the quantitative polymerase chain reaction.

Primer	Sequence	Alignment Temperature	Amplification Efficiency
BAC338F	5′-ACT CCT ACG GGA GGC AG-3′	57 °C	1.99
BAC805R	5′-GAC TAC CAG GGT ATC TAA TCC-3′		
Met630F	5′-GGA TTA GAT ACC CSG GTA GT-3′	57 °C	2
Met803R	5′-GTT GAR TCC AAT TAA ACC GCA-3′		
Oph-151F	5′-GAG CTA ATA CAT GCT AAG GC-3′	55 °C	2
Ento-472R	5′-CCC TCA CTA CAA TCG AGA TTT AAG G-3′		

qPCR: quantitative polymerase chain reaction.

**Table 3 animals-10-00300-t003:** Chemical composition of experimental diets.

Item	Treatments			
	0	1	2	3
Chemical composition (g kg^−1^ DM)				
Organic matter	933	931	925	919
Crude protein	109.7	134.6	136.4	137.8
Neutral detergent fiber	585.6	554.4	537.3	594.2
Acid detergent fiber	295.4	312.1	333.1	352.4
Acid detergent lignin	55.5	74.2	70.4	91.3
Ether extract	11.7	16.3	23.2	22.3
Gross energy (MJ kg^−1^ DM)	17.8	18.0	18.0	18.1
Total phenols ^a^	0.42	9.6	22.8	26.4
Total tannins ^a^	0.04	4.3	13.1	15.1
Condensed tannins ^b^	0.0	2.7	8.2	12.3
In vitro DM digestibility (g kg^−1^ DM)	676	654	637	617
Estimated values				
NFC	22.1	22.1	22.3	16.1

DM: dry matter; ^a^ equivalents-g tannic acid kg DM; ^b^ equivalents-g catechin kg DM; NFC: nonfiber carbohydrate.

**Table 4 animals-10-00300-t004:** Effect of dried *L. leucocephala* leaves supplementation on intake and apparent digestibility in heifers.

Treatments					SE	*p*	Contrast		
Item	0	1	2	3			L	Q	C
	Intake (kg d^−1^)								
DM	8.36	8.32	8.63	8.54	0.13	0.169	0.098	0.8176	0.1226
OM	7.8	7.74	7.98	7.85	0.12	0.340	0.365	0.694	0.136
CP	0.92 ^c^	1.12 ^b^	1.18 ^a^	1.18 ^a^	0.02	<0.0001	<0.0001	0.0004	0.231
NDF	4.9 ^a^	4.61 ^b^	4.64 ^b^	5.07 ^a^	0.08	0.0027	0.065	0.0006	0.695
ADF	2.47 ^d^	2.6 ^c^	2.87 ^b^	3.01 ^a^	0.04	<0.0001	<0.0001	0.882	0.067
GE	149	150	156	155	2.43	0.078	0.0256	0.572	0.181
EE	0.155 ^a^	0.156 ^a^	0.101 ^b^	0.100 ^b^	0.02	0.029	0.008	0.934	0.105
ADL	0.615	0.609	0.631	0.623	0.01	0.114	0.106	0.809	0.057
PT	0.035 ^d^	0.08 ^c^	0.197 ^b^	0.226 ^a^	0.003	<0.0001	<0.0001	0.011	<0.0001
TT	0.003 ^d^	0.036 ^c^	0.113 ^b^	0.129 ^a^	0.002	<0.0001	<0.0001	0.001	<0.0001
CT	0.00 ^d^	0.023 ^c^	0.07 ^b^	0.104 ^a^	0.002	<0.0001	<0.0001	0.003	0.0003
	Nutrient apparent digestibility (g kg^−1^ DMI)								
OM	481.9 ^a^	474.7 ^a^	439.3 ^b^	418.4 ^b^	12.32	0.006	0.001	0.463	0.315
CP	60 ^c^	74.9 ^a^	68.5 ^b^	61.9 ^c^	1.6	0.0003	0.928	<0.0001	0.005
NDF	238.2 ^a^	208.0 ^b^	184.6 ^c^	206.4 ^bc^	9.51	0.008	0.007	0.008	0.247
ADF	79.5	89.9	95.5	85.9	6.02	0.157	0.244	0.057	0.607
DE	8.8 ^a^	8.7 ^a^	8.2 ^ab^	7.8 ^b^	0.25	0.020	0.004	0.360	0.594
	Estimated values								
TDN	0.546	0.541	0.529	0.480					
ME	8.26	8.18	8.00	7.26					

DM: dry matter; DMI: dry matter intake; OM: organic matter; CP: crude protein; NDF: neutral detergent fiber; ADF: acid detergent fiber; GE: gross energy (MJ d^−1^); EE: ether extract; TP: total phenols; TT: total tannins; CT: condensed tannins; ADL: acid detergent lignin; DE: digestible energy (MJ kg^−1^ DMI); TDN: total digestible nutrients (kg^−1^ DMI); ME: metabolizable energy (MJ kg^−1^ DMI). Means in the same row with different superscript letters differ (*p* < 0.05); SE: standard error; surface response: L: linear contrast; Q: quadratic contrast; C: cubic contrast.

**Table 5 animals-10-00300-t005:** Effect of dried *L. leucocephala* leaves supplementation on fermentation in the rumen of heifers.

Treatments					SE	*p*	Contrast		
Item	0	1	2	3			L	Q	C
Rumen pH	6.4	6.5	6.5	6.5	0.13	0.815	0685	0.522	0.604
Total VFA mMol L^−1^	79.9	80.4	75.4	66.9	8.46	0.36	0.128	0.435	0.975
Acetic:propionic acid ratio mMol L^−1^	2.68 ^b^	2.74 ^b^	3.42 ^a^	3.38 ^a^	0.25	0.04	0.013	0.805	0.145
Molar proportions of VFA (%)									
Acetic acid	61.0 ^b^	61.0 ^b^	65.4 ^ab^	67.4 ^a^	1.8	0.025	0.005	0.446	0.298
Propionic acid	22.8	23.1	19.3	20.0	1.7	0.157	0.065	0.871	0.171
Butyric acid	10.9	13.1	10.7	10.8	2.98	0.818	0.772	0.619	0.481
Isobutyric acid	0.62	0.59	1.42	0.42	0.65	0.480	0.920	0.334	0.241
Valeric acid	1.27	1.32	1.69	0.91	0.45	0.464	0.646	0.244	0.350
Isovaleric acid	3.23	0.72	1.36	0.38	1.86	0.479	0.226	0.582	0.449

VFA: Volatile fatty acids. Means in the same row with different superscript letters differ (*p* < 0.05); SE: standard error; surface response: L: linear contrast; Q: quadratic contrast; C: cubic contrast.

**Table 6 animals-10-00300-t006:** Indicators of the flow of nitrogen consumed by heifers fed *Leucaena leucocephala*.

Treatments					SE	*p*	Contrast		
Item	0	1	2	3			L	Q	C
Ratio CP:TDN	0.210	0.249	0.258	0.287					
Ratio TDN:CP	4.98	4.02	3.88	3.48					
N intake (g d^−1^)	146.9 ^b^	179.1 ^a^	188.4 ^a^	188.2 ^a^	3.18	<0.0001	<0.00014	0.0004	0.231
BUN (mg dL^−1^)	9.89 ^c^	10.96 ^b^	11.72 ^a^	9.95 ^c^	0.26	0.001	0.314	<0.001	0.036
N fecal (g d^−1^)	67.4 ^d^	80.1 ^c^	93.7 ^b^	103.8 ^a^	2.4	<0.0001	<0.0001	0.470	0.589
Ratio N fecal:N intake	0.46	0.45	0.50	0.55					
Ratio CP:DOM	0.228	0.284	0.311	0.329					

CP: crude protein; TDN: total digestible nutrients; N: nitrogen; BUN: concentration of blood nitrogen urea; DOM: digestible organic matter; ^a–d^ Means in the same row with different superscript letters differ (*p* < 0.05); SE: standard error; surface response: L: linear contrast; Q: quadratic contrast; C: cubic contrast.

**Table 7 animals-10-00300-t007:** Estimation of ruminal microbial population by quantitative polymerase chain reaction.

Treatments					SE	*p*	Contrast		
Item	0	1	2	3			L	Q	C
qPCR Microbial population									
Protozoa log_10_ cell ml^−1^	4.52	4.77	5.31	5.06	0.34	0.209	0.092	0.329	0.353
Bacteria log_10_ cell ml^−1^	9.01	9.29	9.40	9.22	0.13	0.133	0.147	0.058	0.812
Methanogens log_10_ cell ml^−1^	6.40	6.32	6.77	6.71	0.29	0.395	0.182	0.963	0.304
Methanogen:bacteria	0.71	0.68	0.72	0.73	0.035	0.556	0.433	0.459	0.368

qPCR: quantitative polymerase chain reaction. Means in the same row with different superscript letters differ (*p* < 0.05); SE: standard error; surface response: L: linear contrast; Q: quadratic contrast; C: cubic contrast

**Table 8 animals-10-00300-t008:** Effect of supplements of dried *L. leucocephala* leaves on enteric CH_4_ production in heifers.

Treatments					SE	*p*	Contrast		
Item	0	1	2	3			L	Q	C
Methane									
CH_4_ (g d^−1^)	174.2 ^a^	162.9 ^b^	154.8 ^b^	140.0 ^c^	3.73	0.001	0.0002	0.581	0.497
CH_4_ (DMI)	20.8 ^a^	19.6 ^a^	17.9 ^b^	16.4 ^c^	0.57	0.0012	0.0002	0.672	0.716
Methane g kg^−1^ of digestible fractions intake									
CH_4_ (DM)	44.8	42.5	40.6	39.0	2.37	0.181	0.040	0.860	0.994
CH_4_ (OM)	44.1	42.1	40.8	39.4	2.19	0.274	0.068	0.869	0.910
CH_4_ (CP)	353.4 ^a^	265.8 ^b^	261.9 ^b^	266.1 ^b^	13.68	0.001	0.0009	0.0032	0.131
CH_4_ (NDF)	90.4	98.4	97.1	80.1	6.47	0.097	0.169	0.034	0.767
Estimated of methane as energy loss									
Ym	6.5 ^a^	6.07 ^b^	5.54 ^c^	5.05 ^d^	0.17	0.0008	0.0001	0.841	0.827
CH_4_ (MJ d^−1^)	9.6 ^a^	9.0 ^b^	8.6 ^b^	7.7 ^c^	0.23	0.001	0.0002	0.581	0.497
EF	63.5 ^a^	59.4 ^b^	56.5 ^b^	51.1 ^c^	1.56	0.001	0.0002	0.581	0.497

CH_4_: methane: CH_4_d: CH_4_ g d^−1^; CH_4_DMI: CH_4_ g kg^−1^ Dry Matter Intake; CH_4_DM: CH_4_ g kg^−1^ Dry Matter; CH_4_OM: CH_4_ g kg^−1^ Organic Matter; CH_4_PC: CH_4_ g kg^−1^ Crude Protein; CH_4_NDF: CH_4_ g kg^−1^ Neutral Detergent Fiber; GEI: gross energy intake; Ym: CH_4_ MJ d^−1^, expressed as percentage of gross energy intake; EF: CH_4_ emission factor, kg CH_4_ animal^−1^ year^−1^; Means in the same row with different superscript letters differ (*p* < 0.05); SE: standard error; surface response: L: linear contrast; Q: quadratic contrast; C: cubic contrast.

**Table 9 animals-10-00300-t009:** Estimate global warming potential in heifers fed with dried *L. leucocephala* leaves.

Treatments				
Global warming potential	0	1	2	3
CH_4_ (CO_2_-eq kg d^−1^)	4.88	4.56	4.33	3.92
N_2_O (CO_2_-eq kg d^−1^)	1.17	1.39	1.62	1.79
Total global warming potential				
CO_2_-eq kg d^−1^	6.05	5.95	5.95	5.71

CH_4_; methane; N_2_O nitrous oxide; CO_2_-eq; carbon dioxide equivalents.

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
