# Peer review of "Effect of Dried Leaves of *Leucaena leucocephala* on Rumen Fermentation, Rumen Microbial Population, and Enteric Methane Production in Crossbred Heifers"

_animals, 2020, doi:10.3390/ani10020300_

Round 1
Reviewer 1 Report
The article is interesting and clearly presented.
I have the following minor (mostly editorial) comments:
Line 106 ".....[]."
Line 335 "....ration...."
Line 338 "....effect..."
Attention to the references, they should be described as indicated in the Istruction for the Authors "
Journal Articles:1. Author 1, A.B.; Author 2, C.D. Title of the article. Abbreviated Journal Name Year, Volume, page range. Books and Book Chapters:
2. Author 1, A.; Author 2, B. Book Title, 3rd ed.; Publisher: Publisher Location, Country, Year; pp. 154–196.
3. Author 1, A.; Author 2, B. Title of the chapter. In Book Title, 2nd ed.; Editor 1, A., Editor 2, B., Eds.; Publisher: Publisher Location, Country, Year; Volume 3, pp. 154–196.
Author Response
we thank for the comments
Line 106 ".....[]." The correction was made
Line 335 "....ration...." The correction was made
Line 338 "....effect..." The correction was made
Attention to the references, they should be described as indicated in the Instruction for the Authors " The correction was made.

Reviewer 2 Report
I read with interest your manuscript.
I have some general comments and then some specific edits.
Introduction
The comment about GHG being 14.5% from livestock has been changed it is now around 4.5% or so. Please see the correction which was put forth to FAO by Dr. Mitloehner at UC davis as this has been corrected.
96 Please use the English names for the crops used, latin is not necessary.
127 When exactly was blood sampled?
137 Where is Agilent technologies?
139 Woonsocket, RI, USA
Why is Table 9 here? likewise Table 8? shouldn't table 8 be table 1 this is confusing
Table 8 ? Use common names not latin
What do the superscripts mean?
Could the response be due to differences in diets not just the CT?
186 where is Sable systems?
208-216 why glm and not mixed?
Table 1??
231 Don't repeat data in text that refers to data in a table refer to the table
243 not what your table states
All data tables please insert p values as that will help the reader I suggest avoiding NS you have the values please add
Table 3 insert p values
also was pH changed to H ion before analyzing that is the correct way to analyze pH. It is not appropriate to analyze H ion as pH. Many researchers do that but it is incorrect. Please see M.R. Murphy I believe it is 1981 or 1982 in the Journal of Dairy Science.
264 is not true according to your table 4 data, you indicate differences.
Where are the SE's?
Table 5 p values?
Table 6 where are the rest of the contrasts?
Table 7 statistics?
318 mitigation
327 I may be wrong but I have never heard of a histatin or a statherine.
340-341 Do you mean NDF?
390- 392 can't say this as data are similar not statistically different.
420-422 what is PC?
Author Response
I have some general comments and then some specific edits.
Introduction
The comment about GHG being 14.5% from livestock has been changed it is now around 4.5% or so. Please see the correction which was put forth to FAO by Dr. Mitloehner at UC davis as this has been corrected. We appreciate the comment, however we believe that the suggested change does not apply in the document. derived from the fact that the Doctor's data refers to All of animal agriculture contributes 3.9 percent of total U.S.A. greenhouse gas emissions(Pitesky et al., 2009), while the information in the introduction refers to 14.5% of greenhouse gas emissions worldwide.
96 Please use the English names for the crops used, latin is not necessary. The correction was made.
127 When exactly was blood sampled? Blood samples were taken from animals on days 14 and 15 of every period, within four hours postprandial.
137 Where is Agilent technologies? Santa Clara, CA, United States
139 Woonsocket, Rhode Island, USA
Why is Table 9 here? likewise Table 8? shouldn't table 8 be table 1 this is confusing The number of tables was correcte
Table 8 ? Use common names not latin. The correction was made
What do the superscripts mean? the super indices were removed and the physical form of of each ingredient is described on the table.
Could the response be due to differences in diets not just the CT?
We do believe that the results shown in the present work are derived from the type of protein in the ingredients. The type of protein is related to the type of amino acids and the terciary and quaternary structure of proteins, which affects the extent of protein degradability by microbial enzymes. That is why, such differences result in the lower or greater degree of protein degradation in the rumen. In the case of the employed ingredients in this experiment, soybean meal has a quickly degradable protein fraction (QDP) of 8.9%; a slowly degradable protein fraction (SDP) of 87.2%, and an undegradable protein fraction (UPF) of 3.9% (Lee et al. 2016). It is in this context, that Leucaena leucocephala has a degradable protein fraction of 54.43% and an undegradable protein fraction of 45.57% (Miranda et al. 2012). It may be also that condensed tannins in the leaves of Leucaena leucocephala may be involved in the undegradable protein fraction, thus we could not separate the effect of type of protein and condensed tannins in the present work.
On the other hand, in the case of the decreasing effect on apparent digestibility of crude protein by the inclusion of condensed tannins, it has been demonstrated that condensed tannins forms complexes among them and endogenous protein in the intestine, which results in an increase in introgen in feces (Kariuki and Norton, 2008; Mueller-Harvey, 2006).
It has also been reported a linear reduction in OM digestibility with the increase in the doses of condensed tannins (Makkar, 2003). In the present work, OM digestibility and NDF digestibility decreased with incorporation of L. leucocephala. Maybe this reduction in NDF digestibility resulted from the formation of complexes between condensed tannins with cellulose, hemicellulose, as well as with enzymes of microbial origin which constrain fermentation and digestibility of the fibre fractions (Makkar, 2003; McSweeney et al., 2001). In an experiment carried out by Knapp et al. (2014) it was shown that the content of digestible NDF is the best index of methane production in the rumen, compared with the total content of NDF in the ration, since NDF is the main substrate fermented in the rumen which yields methane. It is because of that, that we believe that the reduction in methane emissions in the present work was the result of the condensed tannins contained in the leaves of L. leucocephala.
186 where is Sable systems? Las Vegas, NV, USA.
208-216 why glm and not mixed?
The latin square design assumes that there is no interaction between rows, columns, and treatments (Montgomery, 2017;). Based on this, it was asumed that all effects were fixed, so the procedure selected for the analysis of this experiment was Proc GLM. However, echoing the suggestion by the reviewer, we revised the literature to make this point clear. Latin square designs can be analyzed either by Proc GLM or with mixed procedures. In the event of using Proc Mixed the literature recommends using this mainly in the case of using two treatments which may be interacting among themselves (Montgomery, 2019). Proc Mixed can also be employed in the following design: imcomplete blocks (Cheng et al., 2005); block should be considered random when the inference is going to be extended to the herd, the treat cow as a random blocking factor; Sometimes measuring same EU over time/period also recommends using a random blocking (Montgomery, 2019).
Based on the statistical literature reviewed, we carried out a Proc Mixed. The random blocking factor was the cow, however, the results obtained with Proc Mixed were no different from those obtained previously with Proc GLM for most of the response variables
Table 1? The number of tables was corrected
231 Don't repeat data in text that refers to data in a table refer to the table The correction was made.
243 not what your table states The correction was made.
All data tables please insert p values as that will help the reader I suggest avoiding NS you have the values please add. The correction was made
Table 3 insert p values. The correction was made
also was pH changed to H ion before analyzing that is the correct way to analyze pH. It is not appropriate to analyze H ion as pH. Many researchers do that but it is incorrect. Please see M.R. Murphy I believe it is 1981 or 1982 in the Journal of Dairy Science.
264 is not true according to your table 4 data, you indicate differences. The correction was made
Where are the SE's? SE is standard error of the mean
Table 5 p values? The correction was made
Table 6 where are the rest of the contrasts? In the case of Ration CP:TDN, TDN:CP, Facial:N intake and CP:DOM was calculated with mean valor for every ration, therefore these rations do not have statistics valor for p or contrast.
Table 7 statistics? The correction was made
318 mitigation. The correction was made
327 I may be wrong but I have never heard of a histatin or a statherine.
These proteins are in the saliva of humans, therefore were eliminated.
Statherin is a protein in humans. It prevents the precipitation of calcium phosphate in saliva, maintaining a high calcium level in saliva available for remineralisation of tooth enamel and high phosphate levels for buffering.
Histatins are histidine-rich (cationic) antimicrobial proteins found in saliva.
340-341 Do you mean NDF? the abbreviation was corrected to NDF
390- 392 can't say this as data are similar not statistically different. The correction was made
420-422 what is PC? the abbreviation was corrected to CP.
we appreciate your comments

Reviewer 3 Report
The Manuscript entitled "Effect of Dried Leaves of leucaena leucocephala on Rumen Fermentation, Rumen Microbial Population and Enteric Methane Production in Crossbred Heifers" (ID Animaks_681943) is an original paper aimed to study the inclusion of Leucaena leucocephala in heifers’ diets. The subject is in line with the objectives of the journal. In my opinion it is acceptable for publication in the Animals journals after some minor revisions.
General Comments:
I think that the content of the work is an interesting contribution to the field of investigation and useful to deep the knowledge about the use of Leucaena leucocephala for heifers. In particular, the aim of study is to evaluate in vivo the effect of dietary supplementation of heifers with increasing levels of dried Leucaena leucocephala leaves on nutrient digestibility, fermentation parameters, microbial rumen population and production of enteric methane.
In my opinion, the Introduction section is complete. In Material and Methods section all the description of the procedures can be reduced. The Results are clear, and the Discussions are well treated. The results of the investigation should be also discussed considering the results on Leucaena leucocephala obtained in vitro by Musco N, Koura I, Tudisco R, et al. Nutritional Characteristics of Forage Grown in South of Benin Asian-Australas J Anim Sci 2016;29(1):51-61. DOI: https://doi.org/10.5713/ajas.15.0200). The Conclusion summarize correctly the results obtained.
Specific comments:
L1: in the title use capital letter for ‘Leucaena’
L17: add ‘dietary’ before supplementation
L30: delete ‘digestible’ before energy (you are speaking about digestibility…)
L38: The authors could select keywords that are not included in the title (i.e. volatile fatty acids, digestibility)
L41-42: move ‘potential’ before ‘global’
L53: ‘(L. leucocephala)’ not necessary
L59: replace ‘; complexes that’ with ‘and’
L85: ‘(DLL)’ not necessary
L91: correct ‘Table 1’ (also L96)
L153 and L160: change Table 9 to Table 2
L212-216: change the sentence (more correctly) ‘The statistical model was Yijkl = μ + Pi + Aj + Tk + Eijk; where: Y is the dependent variable, μ is the general mean, P is the effect of period, A is the effect of animal, T is the effect of treatment and E is the residual error.’
L219: correct Table 1 in Table 3
L224-226: improve the grammar in the sentence
L229: delete ‘MJ: Megajoule;’ (not necessary)
L232: correct Table 2 in Table 4
L243: in material and methods you talked about gross and no digestible energy…
L257: correct Table 3 in Table 5 (also in the table title)
L275: correct Table 4 in Table 6 (also in the table title)
L258-259: check please the not significant differences in Molar concentrations VFA (especially 0 vs. 1)
L280: format correctly the subtitle ‘3.6. Methane production’
L302: format correctly the subtitle ‘3.7.’
L310-315: delete, not necessary, already reported in Material and Methods
L331: delete ‘(Table 1)’ (also elsewhere no table citation in Discussion section)
L314: correct the format for the citation [8]
L495: delete coma after parenthesis
Table
Table 1 (no 8): I wouldn’t use the Latin name for the ingredients. At least only for L. leucocephala
The tables’ numeration needs to be corrected in all the manuscript
References
Maybe the citations are too many.
Check the format for references No. 2 (authors), 19 (year) and 44 (year).
Author Response
In my opinion, the Introduction section is complete. In Material and Methods section all the description of the procedures can be reduced.
The methods in Quantitative real-time polymerase chain reaction and Methane production were reduced
The Results are clear, and the Discussions are well treated. The results of the investigation should be also discussed considering the results on Leucaena leucocephala obtained in vitro by Musco N, Koura I, Tudisco R, et al. Nutritional Characteristics of Forage Grown in South of Benin Asian-Australas J Anim Sci 2016;29(1):51-61. DOI: https://doi.org/10.5713/ajas.15.0200). The Conclusion summarize correctly the results obtained.
Musco et al., 2019 was included.
Specific comments:
L1: in the title use capital letter for ‘Leucaena’ The correction was made
L17: add ‘dietary’ before supplementation The correction was made
L30: delete ‘digestible’ before energy (you are speaking about digestibility…) The correction was made
L38: The authors could select keywords that are not included in the title (i.e. volatile fatty acids, digestibility) The correction was made
L41-42: move ‘potential’ before ‘global’ The correction was made
L53: ‘(L. leucocephala)’ not necessary The correction was made
L59: replace ‘; complexes that’ with ‘and’ The correction was made
L85: ‘(DLL)’ not necessary The correction was made
L91: correct ‘Table 1’ (also L96) The correction was made
L153 and L160: change Table 9 to Table 2 The correction was made
L212-216: change the sentence (more correctly) ‘The statistical model was Yijkl = μ + Pi + Aj + Tk + Eijk; where: Y is the dependent variable, μ is the general mean, P is the effect of period, A is the effect of animal, T is the effect of treatment and E is the residual error.’ The correction was made
L219: correct Table 1 in Table 3. The correction was made
L224-226: improve the grammar in the sentence The correction was made
L229: delete ‘MJ: Megajoule;’ (not necessary) The correction was made
L232: correct Table 2 in Table 4 The correction was made
L243: in material and methods you talked about gross and no digestible energy… The lines 107 to 109 describes how digestible energy was calculated.
L257: correct Table 3 in Table 5 (also in the table title) The correction was made
L275: correct Table 4 in Table 6 (also in the table title) The correction was made
L258-259: check please the not significant differences in Molar concentrations VFA (especially 0 vs. 1) In this case, it was an error of finger, the valor was changed
L280: format correctly the subtitle ‘3.6. Methane production’ The correction was made
L302: format correctly the subtitle ‘3.7.’ The correction was made
L310-315: delete, not necessary, already reported in Material and Methods The correction was made
L331: delete ‘(Table 1)’ (also elsewhere no table citation in Discussion section) The correction was made
L314: correct the format for the citation [8] The correction was made
L495: delete coma after parenthesis The correction was made
Table
Table 1 (no 8): I wouldn’t use the Latin name for the ingredients. At least only for L. leucocephala d The correction was made
The tables’ numeration needs to be corrected in all the manuscript. The correction was made
References
Check the format for references No. 2 (authors), 19 (year) and 44 (year). The correction was made.
we appreciate your comments
